# Natural Products from Marine Microorganisms with Agricultural Applications

**DOI:** 10.3390/md23110438

**Published:** 2025-11-14

**Authors:** Michi Yao, Hafiz Muhammad Usama Shaheen, Chen Zuo, Yue Xiong, Bo He, Yonghao Ye, Wei Yan

**Affiliations:** 1The Sanya Institute, Nanjing Agricultural University, Sanya 572000, China; 2025202097@stu.njau.edu.cn (M.Y.); 2023202085@stu.njau.edu.cn (H.M.U.S.); 2021102129@stu.njau.edu.cn (C.Z.); 2023102132@stu.njau.edu.cn (Y.X.); hebo@njau.edu.cn (B.H.); yeyh@njau.edu.cn (Y.Y.); 2State Key Laboratory of Agricultural and Forestry Biosecurity, State & Local Joint Engineering Research Center of Green Pesticide Invention and Application, College of Plant Protection, Nanjing Agricultural University, Nanjing 210095, China

**Keywords:** natural products, marine microorganisms, agricultural application, pesticides

## Abstract

Global agricultural production is challenging due to climate change and a number of phyto-pathogenic organisms and pests that pose a significant threat to both crop growth and productivity. The growing resistance of pests and diseases to synthetic chemicals makes crop production even more difficult, which highlights the urgent need for alternative solutions. From this perspective, marine microorganisms have emerged as a significant natural product source for their distinctive bioactive compounds and environmentally sustainable potential pesticidal activity. The unique microbial resources and structurally diverse metabolites of the marine ecosystem have been proven to have strong antagonistic effects against a broad spectrum of agricultural diseases and pests, making them a valuable candidate for the development of novel pesticides. This review highlights 126 marine natural products from marine microorganisms with diverse metabolic pathways and bioactivities against agricultural pests, pathogens, and weeds. The findings underscore the potential of marine-derived compounds in addressing the growing challenges of crop protection and offering an appealing strategy for future agrochemical research and development.

## 1. Introduction

Crop cultivation has long been an integral part of human civilization that has contributed to both economic development and population growth. The world’s population is expected to increase substantially from 8 billion to 9.7 billion people by 2050, posing a significant planning challenge for food security [1]. Therefore, sufficient crop production must be ensured to sustain human survival and social stability. The United Nations (UN) has made ending world hunger and poverty a priority in its 2030 sustainable development goals (SDGs) agenda. Unfortunately, global crop growth and production has been extensively influenced by several biotic and abiotic factors. Among biotic factors, disease causative agents such as fungi [2], bacteria [3], and viruses, as well as weeds and pests, have become the potential threats towards successful agricultural production. No doubt, the use of synthetic pesticide is the most powerful approach in controlling plant disease and achieving high-quality yields. However, the excessive use of these chemical substances has led to serious problems for the environment, human health, water, soil fertility, emergence of pathogen resistance, and targeting several beneficial species [4,5]. Therefore, scientists have diverted their attention towards several ecofriendly management strategies, such as safer pesticides, while keeping in view rising resistance levels and the growing demand for safer and more effective drugs and pesticides. The discovery of new lead compounds with high activity, good selectivity, and novel targets or mechanisms of action presents a major opportunity and challenge in technological innovation. The search for new pesticide lead compounds with novel structural frameworks has become a key focus of current research.

Natural products refer to compounds synthesized or extracted from organisms in nature, typically secondary metabolites that exist in plants, microorganisms, animals, or their secretions [6]. Researchers worldwide have screened and isolated natural products from plants, microorganisms, and marine organisms, and evaluated the bioactivities of these compounds. This has led to the discovery of valuable compounds, including some of the most effective agrochemicals in history [7], such as antifungal agents (e.g., Nisin), antiparasitic agents [8,9,10] (e.g., avermectins), insecticides [11] (e.g., spinosad), and herbicides (e.g., glyphosate). The discovery of new active ingredients heavily relies on natural products, particularly in the field of insecticides, where 70% of the insecticides available worldwide are natural products or their derivatives. Similarly, approximately 60% of herbicides and fungicides are based on natural compounds [12]. The marine environment is the largest and most biodiverse ecosystem on Earth, with abundant microbial resources. These species exhibit unique growth environments and are rich sources of novel bioactivities and metabolic pathways, with diversified structures that distinguish them from terrestrial microbes. Researchers are increasingly interested because of their capability of producing structurally unique compounds [13].

Marine natural products have potential inhibitory effects towards several harmful organisms. These microorganisms have emerged as promising sources for discovering pesticides, with their strong antagonistic activity against several agricultural pathogens, pests and weeds. There is strong evidence from advances in science and technology that microorganisms once again play a key role. In the future, several interesting and novel compounds with potential effects will likely to be discovered from the ocean.

This review introduces 126 natural products from marine microorganisms with various structural types and bioactivity against agricultural pathogens, pests, or weeds and a total of 105 references are cited.

## 2. Results

Several types of phyto-pathogens such as *Fusarium* sp., *Alternaria* sp., *Colletotrichum* sp., *Phytophthora* sp., and many others cause significant losses to agricultural production [12,13]. For instance, *Fusarium* sp. is associated with severe pathogenic diseases like *Fusarium* head blight, *Fusarium* ear rot, and *Fusarium* wilt, whereas *Alternaria* sp. is responsible for *Alternaria* wilt, leaf spot, early blight, and leaf blight [14,15]. The natural products from marine microorganisms including fungi and bacteria from marine and mangrove sources have distinct structural characteristics and have shown inhibitory effects against a wide range of pathogens. These antimicrobial compounds have the potential to serve as leading compounds in the development of new pesticides, providing effective, environmentally friendly, and long-lasting solutions for controlling and preventing agricultural pathogens.

### 2.1. Agricultural Antimicrobial Metabolites

#### 2.1.1. Marine-Sourced Fungi (Excluding Those from Mangroves)

*Cladosporium* sp., was isolated from the brown algae *Feldmannia elachistaeformis* collected from Haikou Bay, Hainan, China. The strain produced a novel isochromanone, cladosporinisochromanone (**1**), together with 15 known compounds. Cytochalasin H (**2**) exhibited excellent antifungal activity against *Colletotrichum* sp., with a minimum inhibitory concentration (MIC) of 16 μg/mL, surpassing the MIC value of 64 μg/mL of thiophanate methyl fungicide [16].

Zou et al., conducted a bioassay-guided investigation of a fermentation culture of *Scheffersomyces spartinae* W9, which was isolated from a tropical marine sea. This leads to the production of volatile organic compounds (VOCs), which inhibited the mycelial growth of *Botrytis cinerea* and spore germination, with inhibition rates of 77.8% and 58.3% as compared to a control. Additionally, these VOCs reduced the incidence of disease of *Botrytis cinerea* on the surface of strawberries and the diameter of lesions by 20.7% and 67.4%, respectively, as compared to CK. A series of compounds were isolated and their structures were characterized in order to describe the components of these VOCs. Among them, 2-phenylethanol (**3**) was found to have the strongest antifungal activity for controlling *Botrytis cinerea* in strawberries [17].

In another study, chemical investigations of secondary metabolites from *Aspergillus sydowii* LW09 resulted to the discovery of a wide range of compounds which showed a potential response against agricultural plant pathogens. (7S,11S)-(+)-12-hydroxysydonic acid (**5**) exhibited antagonistic activity against *Pseudomonas syringae*, with an MIC of 32 μg/mL. Aspergillusene B (**4**) and expansol G (**6**) also showed antifungal activity against *Ralstonia solanacarum*, both with MIC values of 32 μg/mL. Additionally, (S)-sydonic acid (**7**) demonstrated excellent inhibitory effects against *Fusarium oxysporum*, with an EC_50_ value of 1.85 μg/mL. Furthermore, compounds **4**–**7** and (−)-(R)-cyclo-hydroxysydonic acid (**8**) were able to inhibit the germination of *Alternaria alternata* [18].

*Penicillium* sp. CRM1540, isolated from Antarctic marine sediments, yielded a bioactive secondary metabolite called cyclopaldic acid. This compound showed a maximum inhibition rate of 90% against *Macrophomina phaseolina*, *Rhizoctonia solani*, and *Sclerotinia sclerotiorum*, with a concentration of 100 μg/mL. Even at a concentration of 50 μg/mL, the inhibition rate against these pathogenic fungi was over 70% [19]. Bin Liu isolated an actinomycete strain from the intertidal zone that was identified as *Streptomyces cinereoruber*. A metabolite named as aurone (**9**) was discovered from the fermentation product of this strain. Greenhouse experiments revealed that herboxidiene exhibited high fungicidal activity against powdery mildew (*Botrytis cinerea*), cucumber powdery mildew (*Sphaerotheca fuliginea*), and rice blast (*Pyricularia oryzae*) disease [20].

Oppong-Danquah et al. reported that the marine-derived fungus *Cosmospora* sp. exhibits inhibitory activity against plant pathogens through microbial co-culture techniques. Further research on culture extracts of this fungus led to the discovery of sudanones A, E, and D (**10**–**12**) and pseudoanguillosporins (**13**). Soudanone E (**11**) and soudanone D (**12**) exhibited antifungal activity against *M. oryzae* and *Phytophthora infestans*, while compound **13** demonstrated broad-spectrum antimicrobial activity against *Pseudomonas syringae*, *Xanthomonas campestris*, *M. oryzae*, and *P. infestans*, with MIC values of 23.4, 7.4, 3.2, and 0.8 μg/mL, respectively [21].

The marine-derived fungus *Aspergillus tabacinus* yielded a number of benzyl ether compounds. MIC values for violaceol I (**14**), violaceol II (**15**), and diorcinol (**16**) ranging from 6.3 to 200 μg/mL have demonstrated broad-spectrum antibacterial activity. Above all, diorcinol (**16**) showed excellent antifungal potency against rice blast (*M. oryzae*), tomato late blight (*P. infestans*), and pepper anthracnose (*C. coccodes*) [22].

Similarly, compounds were obtained from ethyl acetate and n-butanol extracts of the fermentation broth of the marine-derived fungus *Trichoderma longibrachiatum*, which effectively inhibited the growth of tomato gray mold (*B. cinerea*), rice blast (*M. oryzae*), and tomato late blight (*P. infestans*). In vitro tests revealed that all compounds successfully inhibited the fungal growth of *P. infestans*, while bisvertinolone (**17**) showed the strongest inhibitory activity, with an MIC value of 6.3 μg/mL. Moreover, compound **17** exhibited broad-spectrum antimicrobial activity against *C. coccodes*, *Cylindrocarpon destructans*, and *M. oryzae* at concentrations of 12.5, 25, and 50 μg/mL, respectively [23].

The endophytic marine-derived fungus *Aspergillus* sp. SCAU150 provided unsaturated acid derivatives and polyketide compounds, while the structures of these compounds were determined through chemical analysis techniques. Experimental tests revealed that 8-ethyl 2-methyleneoctanedioate (**18**) exhibited inhibitory effects against *Fusarium solani* bio-80814, with a 6 mm diameter inhibitory zone under 5 µg/disc [24].

Similarly, marine-derived fungus *Penicillium* sp. N-5 was the source of three andrastin-type meroterpenoids hemiacetalmeroterpenoids A-C (**19**–**21**), and a drimane sesquiterpenoid astellolide Q (**22**), with eleven known compounds. Parasiticolide A (**23**) was created for the first time from a natural source. An in vitro study revealed that hemiacetalmeroterpenoids A (**19**), citreohybridone A (**24**), and andrastins B (**25**) exhibited significant activity against *Penicillium italicum* and *Colletrichum gloeosporioides*, with MIC values ranging from 1.56 to 6.25 μg/mL [25]. The structures of compounds **1**–**25** are shown in Figure 1.

Saad et al. discovered a growth-inhibitory compound from the secondary metabolites of the marine-derived fungus *Eurotium rubrum* that is effective against agricultural plant pathogens. The compound was identified as methyleurotinone (**26**). It exhibited strong antibacterial activity against *Pectobacterium carotovorum* subsp. *carotovorum*, *Pseudomonas syringae* pv. *syringae*, *Rhizobium radiobacter*, and *Ralstonia solanacearum*, with MIC values of 31.3, 125, 31.3, and 125 mg/L, respectively [26].

Four new compounds were discovered from the strain *Aspergillus terreus* BCC5799, which was isolated from soil endophytic sources in a marine environment. These compounds were identified as asperteramide (**27**), aspulvinone O (**28**), luteoride E (**29**), and asterriquinone CT5 (**30**), with significant antagonistic properties. A bioassay study showed that these compounds exhibited potential inhibitory activity against *Alternaria brassicicola* with MIC values of 6.3, 50.0, 50.0, and 50.0 μg/mL, respectively [27].

*Trichoderma* sp. Z43 isolated from marine brown algae *Dictyopteris* was the source of six new lipids along with a known compound triharzianin B (**33**). These seven compounds were further evaluated for their activity against three plant pathogens. Results revealed that trichoderols B (**31**), trichoderols E (**32**), and triharzianin B (**33**) exhibited inhibitory activity against *Fusarium graminearum*, *Gaeumannomyces graminis*, and *Glomerella cingulata*, with MIC values of 64 μg/mL. Additionally, these compounds showed low toxicity to the brine shrimp *Artermia salina*, indicating their potential as environmentally friendly fungicides [28].

A novel polyketide-terpenoid hybrid natural product, tennessenoid A (**34**), was obtained from *Asperigills* sp. strain 1022LEF, which was derived from marine red alga *Grateloupia turuturu* in Qingdao, China. This compound significantly responds against the growth of several phyto-pathogens, with inhibition zone diameters ranging from 2 to 7 mm [29].

*Penicillium chrysogenum* LD-201810 yielded a pair of new nor-bisabolane enantiomers methylsulfinyl-1-hydroxyboivinianin A (**35**) and two new hydroxyphenylacetic acid derivatives chrysoalide A-B (**37** and **38**), along with four known compounds which were isolated from the culture filtrate. Among them, hydroxysydonic acid (**36**) exhibited excellent antifungal effects against *Botrytis cinerea*, with an EC_50_ value of 13.6 μg/mL [30].

Similarly, five different fungal strains were isolated from sea anemones and these were assessed for their antagonistic properties. *Emericella* sp. SMA01 showed strong activity among five strains of fungi and analysis from the ethyl acetate extract of the fermentation broth revealed its main metabolite as phenazine-1-carboxylic acid (**39**). This compound exhibited inhibitory activity against *Phytophthora capsici*, *Gibberella zeae*, and *Verticillium dahliae*, with EC_50_ values ranging from 23.26 to 53.89 μg/mL [31].

Investigation of red algae endophytic fungi *Trichoderma brevicompactum* A-DL-9-2 resulted in the isolation of a series of trichothecene derivatives. Testing the growth inhibition effects of these compounds on common plant pathogenic fungi revealed that trichodermin (**40**) exhibited the highest activity. The MIC values for compound **40** were 4.0 μg/mL against both *Botrytis cinerea* and *Fusarium oxysporum*, and 8.0 μg/mL against *Phomopsis asparagi* [32].

Three new compounds, polyhydroxy steroid (**41**), tricyclic diterpenoid (**42**) and isaridin, were obtained from a marine-derived entomopathogenic fungus *Beauveria feline*. Biological activity evaluation showed that compounds **41** and **42** exhibited antifungal activity against drug-resistant *Botrytis cinerea*, with MIC values ranging from 16 to 32 μg/mL, significantly outperforming carbendazim (MIC value of 256 μg/mL) [33].

A new meroterpenoid-type alkaloid, oxalicine C (**43**), and two new erythritol derivatives, penicierythritols A (**44**) and B (**45**), were produced by *Penicillium chrysogenum* XNM-12, which was isolated from the marine algae. Among them, compounds **43** and **45** exhibited moderate activity against *Ralstonia solanacearum*, with MIC values of 8 and 4 μg/mL, respectively. Additionally, penicierythritols B (**45**) also showed moderate activity against *Alternaria alternata*, with an MIC value of 8 μg/mL [34].

A co-culture of marine red algal-derived endophytic fungi *Aspergillus terreus* EN-539 and *Paecilomyces lilacinus* EN-531 resulted in the production of a new terrein derivative, asperterrein (**46**) with two known compounds dihydroterrein (**47**) and terrein (**48**). These compounds were not detected when the strains were cultured separately. Antibacterial activity tests revealed that compounds **46**–**48** exhibited inhibitory effects against *Alternaria brassicae*, with MIC values of 64, 64, and 8 μg/mL, respectively, in comparison to control (chloramphenicol) (4 μg/mL) [35].

Three new sesquiterpenoid compounds, chermesiterpenoids A–C (**49**–**51**), were isolated and identified from the secondary metabolites of the endophytic fungus *Penicillium chermesinum* EN-480, derived from red algae. Biological activity tests showed that these compounds exhibited modest inhibitory activity against *Colletottichum gloeosporioides*, with concentrations of 64, 32, and 16 μg/mL, respectively [36]. The structures of compounds **26**–**51** are shown in Figure 2.

#### 2.1.2. Marine-Sourced Bacteria

The ethyl acetate extracts of the fermentation broth of two bacterial strains (*Bacillus velezensis* GP521A and *Pseudoalteromonas caenipelagi* GP3R5) from the marine soil successfully inhibited the mycelial growth of *Colletotrichum camelliae*, *C. fructicola* CF-1, and *Pyricularia oryzae* P131 at concentrations of 100 and 200 μg/mL. Moreover, the extracts also suppressed the conidial germination and appressorial formation of *Colletotrichum* spp., with 50 μg/mL of concentration. Interestingly, the detached oil tea leaves treated with the application of these extracts at 100 μg/mL results in decreasing the infection area significantly by 98.0% and 97.5% [37]. Two new pyrrolizine alkaloids phenopyrrolizins A (**52**) and B (**53**) were identified from the fermentation broth of marine bacterium *Micromonospora* sp. HU138. These two compounds were found inhibiting the fungal growth of *Botrytis cinerea* by 18.9% and 35.9%, respectively, under in vitro conditions [38].

According to NMR and mass spectrometry results, another compound was identified as 2, 4-di-tert butyl phenol (2, 4-DTBP) (**54**), which was isolated from marine bacterial strain *Serratia marcescens* BKACT. This compound potentially inhibited the spore germination growth of *Fusarium foetens* NCIM 1330 at 0.53 mM of concentration. However, it totally suppressed the germination of *Fusarium* sp. spores on wheat seeds without any toxic effects under greenhouse conditions, proving to be a promising candidate for anti-*Fusarium* applications [39].

Similarly, another study revealed that ethyl acetate extract fermentation broth boasts a compound that inhibits the growth and spore germination of plant pathogenic fungus *Alternaria alternata*. It was isolated from plant disease suppressive compost containing residues from the fishing industry and peat moss. Chemical structure elucidation revealed it to be a newly reported compound known as arthropeptide B (**55**) [40].

The secondary metabolites of the marine bacterium *Subtilis* subsp. *spizizenii* MC6B-22 revealed the presence of lipopeptide compounds. Subsequently, these compounds were recognized as mycosubtilin upon further identification, which exhibited broad-spectrum antifungal activity against ten plant pathogenic fungi of tropical crops, with MIC values ranging from 25 to 400 μg/mL. The authors believe that these findings hold potential for applications in agricultural biocontrol [41]. Lipopeptide compounds, including fengycin A and surfactin were discovered from the secondary metabolites of the marine strain *Bacillus amyloliquefaciens* HY2-1. These compounds exhibited a broad-spectrum antifungal activity against seven fungal phyto-pathogens. Electron microscopy revealed that these compounds exert their antifungal effects by disrupting the cell membranes of the pathogenic fungi, thereby inhibiting the formation of fungal mycelia and spores [42]. *Kocuria palustris* 19C38A from marine organism was the source of a class of compounds, benzimidazole (**56**), with antifungal activity against *Fusarium oxysporum*. Further studies on the antifungal mechanism showed that these compounds disrupted the cell integrity of *Fusarium oxysporum* and inhibited spore germination [43].

The culture supernatant of *B. amyloliquefaciens* was collected from the Arctic Ocean and yielded an antifungal peptide W1 with a molecular weight of 2.4 kDa. It was considered as a new antifungal peptide derived from a fragment of preprotein translocase subunit YajC. It prevents the growth of *Sclerotinia sclerotiorum* and *Fusarium oxysporum* at concentrations of 140 and 58 μg/mL [44]. A cyclic lipopeptide, C17-fenngycin B, was identified from the metabolites of the deep-sea-derived bacterium *Bacillus subtilis* 2H11. This compound exhibited an 89.8% inhibition rate against *Fusarium solani* growth at a concentration of 0.2 mg/mL through pot experiments [45]. A new strain of *Streptomyces yongxingensis* sp. nov. (JCM 34965) was obtained from a marine soft coral to check its bioactivity. A bioactive compound, niphimycin C (**57**), was isolated from this strain and exhibited strong antifungal activity against banana wilt disease caused by *Fusarium oxysporum* f. sp. *cubense* pathogen, with an EC_50_ value of 1.2 μg/mL. This compound significantly inhibited the growth of pathogenic mycelia and spore germination. Further studies revealed that niphimycin C reduced the activity of key enzymes in the tricarboxylic acid cycle and electron transport chain. The authors suggest that niphimycin C is a promising agrochemical fungicide for controlling fungal diseases [46]. The structures of compounds **52**–**57** are shown in Figure 3.

#### 2.1.3. Fungi from Mangroves

A variety of compounds were isolated from the secondary metabolites of an endophytic fungus, *Alternaria iridiaustralis*, which was isolated from halophytic plants in coastal areas. A series of compounds was discovered, among which alternanone A, B, D, and C (**58**–**61**) exhibited antifungal activity against benomyl-resistant strains of *Botrytis cinerea*, with MIC values ranging from 32 to 128 μg/mL, which were superior compared to the MIC value of 256 μg/mL with benomyl fungicide. This indicated the potential of these compounds as microbial pesticides for disease control [47].

The solid rice culture of the mangrove-derived fungus *Nigrospora* sp. QQYB1 produced 12 new griseofulvin derivatives. Out of these, compounds **62** and **63** demonstrated inhibitory effects against *Colletotrichum truncatum*, *Microsporum gypseum*, and *Trichophyton mentagrophyte* [48].

Similarly, an endophytic mangrove-derived fungus (*Trichoderma lentiforme* ML-P8-2) strain, was the source of a tandyukusin derivative and polyketide compounds. From the culture broth of this strain, compounds **64** and **65** both exhibited antifungal activity against *Penicillium italicum*, with an MIC value of 6.25 μM. Furthermore, biological activity tests indicated that tandyukisin C and G (**66** and **67**) had shown inhibitory action against the *Penicillium italicum* pathogen, both with 12.5 μg/mL concentrations [49].

Two strains of *Penicillium javanicum* HK1-23 as well as *P. janthinellum* HK1-6 were obtained from mangrove rhizosphere soil of the South China Sea. Activity-guided isolation from these two strains led to the discovery of the antibacterial secondary metabolites brefeldin A (**68**) and penicillic acid (**69**). At the same concentration (50 μg/mL), penicillic acid inhibited *Rhizoctonia solani* and *R. cerealis* by 67.5% and 76%, respectively. In contrast, brefeldin A showed a 56.4% inhibition rate against *R. cerealis* [50].

Two novel heterodimeric tetrahydroxanthones (aflaxanthones A (**70**) and B (**71**)), dimerized via an unprecedented 7,7′-linkage, have been obtained from the mangrove endophytic fungus *Aspergillus flavus* QQYZ. Aflaxanthones A (**70**) had an MIC value of 12.5 μM against *Colletotrichum gloeosporioides* and 3.13 μM against *Fusarium oxysporum*. Aflaxanthones B (**71**) also showed MIC values of 12.5 μM against both *Fusarium oxysporum* and *Collettrichum musae* indicating broad-spectrum and potential antifungal activity under in vitro conditions [51].

Investigation of *Cladosporium cladosporioides* MA-299 identified a bicyclic 5/9 ring system, macrolide cladocladosin A (**72**), along with two new sulfur-containing macrolides, thiocladospolides F (**73**) and G (**74**). The antifungal activity tests against *Fusarium oxysporum* f. sp. *momodicae* revealed that these compounds exhibited MIC values of 32, 16, and 32 μg/mL, respectively, as compared to a positive control (amphotericin B) value of 0.5 μg/mL [52].

Several new compounds were isolated from the secondary metabolites of the fungus *Penicillium javanicum* HK1-23, which was sourced from mangrove rhizosphere soil. The inhibitory activity of emindole SB (**75**) and penialidin A (**76**) against *Alternaria alternata* at gradient concentrations of 50.0, 10.0, and 2.0 μg/mL was evaluated. The results showed that emindole SB (**75**) exhibited a higher inhibitory activity at 50 μg/mL, with an inhibition rate of 77.3%, whereas penialidin A (**76**) had an inhibition rate of 56.8% at the same concentration [53].

The endophytic fungus *Penicillium coffeae* MA-314, isolated from the mangrove plant *Laguncularia racemosa*, has produced a new δ-lactone penicoffeazine A (**77**) and pairs of new isocoumarin derivatives (**78** and **79**). Penicoffeazine A (**77**) showed strong antifungal potency against *Fusarium oxysporum* f. sp. *momordicae* nov. f. and *Colletotrichum gloeosporioides*, with an MIC value of 5 μM, which is comparable to the positive control amphotericin B [54].

Four new isocoumarin derivatives, botryospyrones A, B, C, and D (**80**–**83**), as well as a new natural tryptamine derivative compound **84**, were isolated through spectroscopic structure elucidation. These were purified from the culture broth of a novel endophytic fungus, *Botryosphaeria ramose* L29, which was derived from the leaf of *Myoporum bontioides*. These were further assessed for their antagonistic properties and showed MIC values of 25.0, 25.0, 50.0, and 6.2 μg/mL, respectively. Surprisingly, compound **84** was 16 times more potent than the positive control, triadimefon (MIC = 100 μg/mL). Additionally, all compounds, except for compound **80** (MIC = 199 μg/mL), exhibited significant antifungal activity as compared to triadimefon (MIC = 150 μg/mL), whereas compound **84** (MIC = 6.2 μg/mL) was found to be 18 times more active than triadimefon [55].

Study on the secondary metabolites of the marine-derived fungus *Alternaria* sp. (P8) led to the discovery of a new benzopyranone (+)-(2S,3R,4aR)-altenuene (**85**). This compound exhibited inhibitory effects on the growth of *Alternaria brassicicola* [56]. The structures of compounds **58**–**85** are shown in Figure 4.

### 2.2. Agricultural Antiviral Metabolites

Plant viral diseases, often referred to as “plant cancer” [57], lead to significant yield losses once crops are infected. These diseases pose a major threat to agricultural production worldwide due to their complex transmission mechanisms and the challenging management strategies required. The prevention and control of plant viruses mainly rely on preventive and indirect methods [58]. Currently, commercial antiviral agents such as amino-oligosaccharide, ribavirin, and ningnanmycin are available [59,60], but their field control efficacy is less than 60% [61]. Many natural products derived from marine microorganisms have been found to express antiviral activities (including activities against animal viruses). Some researchers have used natural products with antiviral activity as a scaffold to design and synthesize a series of derivatives with enhanced antiviral properties [62].

EI-Gendy et al. discovered the structure of essramycin (**86**), produced by *Streptomyces* Merv 8102. However, its antiviral properties were not examined at that time [63]. As of 2020, this compound was found to have significant inhibitory effects on the tobacco mosaic virus (TMV). Using essramycin (**86**) as the main compound, a series of its derivatives were designed and synthesized. Most of these derivatives exhibited stronger antiviral activity than ribavirin. Furthermore, compounds **87** and **88** even surpassed ningnanmycin in efficacy. Therefore, essramycin (**86**) holds the potential to be the main compound in the development of novel antiviral treatments [64].

A compound named acterophenone A (**89**) was identified from the secondary metabolites of the marine-derived *Streptomyces* sp. KCB-132. Antiviral experiments revealed that acterophenone A (**89**) expressed significant antiviral activity against plant viruses, including TMV, Tomato Mosaic Virus (ToMV), and Cucumber Mosaic Virus (CMV), with more than 70% inhibition rates. This efficacy even surpasses the positive control agent ningnanmycin, highlighting its potential as an effective antiviral agent [65]. The structures of compounds **86**–**89** are shown in Figure 5.

### 2.3. Agricultural Herbicidal Metabolites

Weeds have a significant negative impact on crop growth and productivity in agricultural production. These significantly interfere with crops for light, water, and nutrients, affecting their normal growth and development. Additionally, weeds are the major source providing habitats for plant pathogens and pests, indirectly harming crop health. Moreover, some weeds can secrete allelopathic substances, which directly inhibit crop growth and even lead to crop death [66]. Currently, there are few studies on the herbicidal activity of marine natural products (MNPs), with only a handful of publications available. However, MNPs hold great potential and research value as herbicides.

A study has highlighted 37 strains of *Alternaria* sp. From marine habitats (P8), which underwent further identification and chemical investigation. It provided one new benzopyranone, along with seven known compounds. Activity tests revealed that (+)-(2S,3R,4aR)-altenuene (**90**), (+)-isoaltenuene (**91**), stemphyperylenol (**92**), and alterperylenol (**93**) exhibited significant phytotoxicity, markedly inhibiting the growth of amaranth seeds and leaves [56].

Alkalodi (**94**) was isolated from the secondary metabolites of marine-derived *Alternaria iridiaustralis*, which exhibited potential growth inhibitory effects on *Echinochloa crusgalli* seedlings. At concentrations of 20 and 40 μg/mL, the inhibitory activity reached 90%, surpassing the herbicide acetochlor. Additionally, alkalodi (**94**) also showed inhibitory effects on other weeds such as *Digitaria sanguinalis*, *Portulaca oleracea*, and *Descurainia sophia* [47].

*Penicillum sclerotiorum* HY5 from mangroves yielded seven pairs of azaphilones E/Z isomers and the structures of these compounds were determined using various analytical techniques. Isochromophilone H (**95**), ochlephilone (**96**), and isochromophilone I (**97**) exhibited strong inhibitory effects on the growth of radicle and plumule on *Amaranthus retroflexus* L., with EC_50_ values ranging from 234.87 to 320.84 μM, compared to the positive control glufosinate-ammonium, with EC_50_ values of 555.11 μM (radicle inhibition) and 656.04 μM (plumule inhibition). Additionally, ochlephilone (**96**) and isochromophilone I (**97**) also inhibited the growth of velvetleaf (*Abutilon theophrasti* Medikus), with EC_50_ values ranging from 768.97 to 1201.52 μM. The authors suggest that these compounds provide new leading compounds for the development of marine-derived bioherbicides [67].

Seed germination tests were conducted with the secondary metabolites of the fungus *Eurotium rubrum*. The tested compounds significantly reduced the germination of *Echinochloa crusgalli* seeds at a concentration of 2 mM. Notably, integric acid A (**98**) and brifeldin A (**99**) completely inhibited seed germination. Additionally, compounds **100**–**104** significantly inhibited the root and shoot growth of *E. crusgalli* at the same concentration [26].

Another study showed 449 marine-derived fungal strains that were screened in order to find compounds inhibiting pyruvate phosphate dikinase (PPDK), a potential herbicidal target in C4 plants. Several isolates were found selectively inhibiting PPDK activity. However, during experimental tests, one isolate was purified and identified as unguinol (**105**). This compound inhibited PPDK via a novel mode of action, with an EC_50_ value of 42.3 ± 0.8 μM [68]. The structures of compounds **90**–**105** are shown Figure 6.

### 2.4. Agricultural Insecticidal Metabolites

Pests can directly reduce crop yield and quality by feeding on leaves, stems, fruits, and other parts of plants. For example, aphids suck plant sap, causing slow growth or even the death of the plant. Additionally, some pests serve as vectors for plant pathogens such as powdery mildew, viruses, and bacteria. These pests transmit diseases from one plant to another, leading to outbreaks and spread of plant diseases [69]. This indirect damage is often more severe than the direct feeding damage. Moreover, controlling these pest-borne diseases is more challenging, which can lead to even greater losses. On the other hand, extensive use of chemical pesticides to control pests has harmful effects on non-target organisms, including beneficial insects, soil microorganisms, and birds, disrupting the balance of agroecosystems [70]. Some pests have developed resistance to synthetic chemicals, making future control efforts more difficult. Currently, many highly effective commercial insecticides are primarily composed of chemically synthesized halides, sulfides, and nitrides. These elements are abundant in marine natural products but are rare in terrestrial natural products [71,72]. As natural sources of chemicals, marine natural products offer a competitive advantage in “green chemistry”.

Four new xanthene derivatives, penicixanthenes A–D (**106**–**109**), were isolated from the secondary metabolites of the mangrove endophytic fungus *Pencillium* sp. JY246. The insecticidal activity of these compounds was tested, revealing that penicixanthenes B and C (**107** and **108**) showed inhibitory activity against the growth of Hubner larvae (*Helicoverpa armigera*), with LC_50_ values of 100 and 200 μg/mL, respectively. Additionally, compounds **106**, **108**, and **109** exhibited insecticidal activity against *Culex quinquefasciatus* larvae, with LC_50_ values of 38.5, 11.6, and 20.5 μg/mL, respectively. These xanthene derivatives show potential for development as insecticides [73].

The marine alga-derived endophytic fungus *Acremonium vitellinum* provided three chloramphenicol derivatives compounds, **110**–**112**. Synthomycin (**110**) had the most significant activity against *Helicoverpa armigera*, with an LC_50_ value of 0.56 μg/mL (compared to the positive control, matrine, with an LC_50_ value of 0.24 μg/mL). Compound **111** was developed as a novel, eco-friendly, and safe biopesticide with an improved profile in agrochemicals [74].

The ethyl acetate extract of the fermentation of *Aspergillus fumigatus* JRJ11048, isolated from the leaves of Hainan-endemic mangrove plant *Acrostichum specioum*, exhibited insecticidal activity against *Spodoptera litura*. Seven compounds were extracted from the secondary metabolites of fungus among which aspergide (**113**) and 11-methyl-11-hydroxyldodecanoic acid amide (**114**) were newly identified. Further studies on these individual compounds revealed that aspergide (**113**) demonstrated significant insecticidal activity against *Spodoptera litura* larvae [75].

*Eurotium cristatum* EN-220, an endophytic fungus isolated from the marine alga *Sargassum thunbergia*, yielded four new indole alkaloids, cristatumins A–D (**115**–**118**) and six known compounds. The insecticidal activity of the isolated compounds was evaluated and showed that cristatumins A (**115**) exhibited moderate lethality against brine shrimp [76].

*Aspergillus oryzae* was the source of two new indoloditerpene derivatives, asporyzin A (**119**) and asporyzin B (**120**), and one new indoloditerpene, asporyzin C (**121**), along with three known indoloditerpenes. Asporyzin C (**121**) exhibited potent activities against *E. coli*, with an inhibition diameter of 8.3 mm. The authors suggest that the presence of the indole and tetrahydrofuran units in JBIR-03 (**122**) likely enhances its insecticidal activity. This indole unit structure offers a new avenue for developing novel indoloditerpene insecticides [77]. The structures of compounds **106**–**122** are shown Figure 7. All the mentioned compounds are listed in Table 1.

## 3. Discussion

This paper summarizes a total of 122 microbial secondary metabolites with agricultural activities. Among these, 106 compounds exhibit agriculturally active properties, while 16 compounds show no agricultural activity. There are 71 compounds with antibacterial activity, 4 compounds with antiviral activity, 16 compounds related to herbicidal activity, and 15 compounds with insecticidal activity. These compounds demonstrate a broad spectrum of activity against multiple genera and species of agricultural pathogenic microorganisms, as well as weeds and pests (Figure 8).

The results indicate that among the compounds with antibacterial activity, those derived from marine fungi, marine bacteria, and mangrove-derived fungi account for 51 (59%), 6 (7%), and 30 (34%), respectively. Among the fungal-derived compounds, the majority of active compounds originate from the genera *Penicillium* sp. (25%) and *Aspergillus* sp. (17%) (Figure 9).

### Mechanism of Action of MBCs

The mechanism of marine bioactive compounds is complex (Figure 10). Plant pathogenic bacteria and fungi rely on their cell walls for cellular stability, osmotic pressure, ion exchange mechanism, and metabolism [78]. Marine-derived bioactive compounds target key component cell walls of fungi, such as glucosamine to prevent cell wall growth and synthesis. For example, the microalgal phenolic extracts (MPEs) derived from the microalgae *Spirulina* sp. and *Nanochloropsis* have been reported to degrade the glucosamine structure of *Fusarium graminearum* while reducing its production by 15% [79]. Similarly, chitinase enzyme from marine bacterium *B. pumilus* hydrolyzed chitin in *F. oxysporum* and inhibited its growth [80].

The cell membrane constitutes a lipid bilayer that exhibits semi-permeability, which regulates the bidirectional movement of molecules within and outside the cells of plant pathogens. The ethyl acetate extract from marine *Streptomyces* sp. AMA49 disrupts the cell membrane of *M. oryzae* and suppresses pathogen growth and spore germination [56,81]. Similarly, marine bioactive compounds have been observed inhibiting bacterial cell membrane integrity against several pathogens [82,83,84,85,86]. Marine natural products also target vital metabolic pathways. Fatty acid metabolism, which is vital for appressorium formation and host penetration were also disrupted to prevent infection [87]. Compounds such as haliangicin isolated from marine myxobacteria impede fungal respiration by inhibiting cytochrome bc-1 complex while blocking the electron transport chain, resulting in cell death [88]. MBCs also disrupt bacterial quorum sensing (QS), a cell density-controlled system that is necessary for gene virulence. Compounds such as 2-methyl-N-(2′-phenylethyl)-butanamide inhibit QS in *Burkholderia glumae*, suppressing production of virulence factors and preventing host infection [89,90,91]. In addition, certain marine natural products induce plant immunity, controlling bacterial pathogens indirectly through stimulating host defense mechanisms as opposes to direct antimicrobial activity [92,93,94].

MBCs also exhibited strong antiviral properties against plant viruses through inhibition of multiple stages of the viral cycle, including attachment, penetration, replication, and viral assembly. The essramycin compound from *Streptomyces* sp. inhibits TMV while disrupting viral assembly and interfering with 20S coat protein disk aggregation [64]. Aldidine derivative and laurene sesquiterpenes similarly inhibited TMV disease [62]. Marine actinomycetes like *Streptomyces* yielded secondary metabolites that can directly inactivate plant viruses like TMV and cucumber mosaic virus (CMV) or induce systemic resistance in plants. Sulfated polysaccharides of marine algae prevent viral adsorption and entry by binding to viral particles or host receptors, blocking infection [95]. Similarly, Euphorbia factor L-1, a marine plant-derived terpenoid from *Euphorbia tirucalli*, has shown anti-viral activity against citrus tristeza virus (CTV) by inhibiting viral RNA synthesis and replication [96]. Seaweed phenolics enhance plant growth and stress resistance as well as seed germination, elongation of roots and shoots, and photosynthesis, and protect against drought, salt, and heavy metal stresses through triggering antioxidants while inhibiting pathogens and pests. These also contribute to cell wall strengthening, hormone modification, and improved plant health and yield [97].

MBCs have been found disrupting the number of physiological and biochemical functions of insect pests. Alkaloid and flavonoid derivatives primarily exert their insecticidal effect by inducing cytotoxicity, which directly damages insect cells. Fatty acids and peptides prevent larval growth, feeding behavior, and transformation by disrupting the protein system and nervous system regulation. Moreover, organosulfur compounds and marine-derived steroids cause muscular and cardiac impairment, leading to paralysis or death of insects. Marine insecticidal proteins, such as cry toxins from marine *Bacillus*, cause cell lysis and larval death by binding to specific receptors in the insect midgut [72]. Manzamine alkaloids isolated from marine sponges exhibited the significant insecticidal activity through disrupting the essential physiological processes of insects such as western corn rootworm [98].

Marine sources exert a herbicidal mode of action by inhibiting photosystem II, disrupting the electron transport chain and interfering with chlorophyll activity, thereby preventing the energy production required for growth and resulting in the death of weeds [99]. Similarly, secondary metabolites such as *Asparagopsis armata* interfere with chloroplast electron transport, inducing oxidative stress and cellular damage in weeds. They also suppress carotenoid metabolism, impairing plant resistance against oxidative damage, which weakens weed defense [100]. Inhibition of acetolactate synthase (ALS), essential for branched-chain amino acid production, additionally impedes protein synthesis and inhibits weed development [101]. These targeted mechanisms of action reduce damage to non-target and beneficial microorganisms, offering sustainable alternatives to synthetic herbicides.

## Figures and Tables

**Figure 1 marinedrugs-23-00438-f001:**
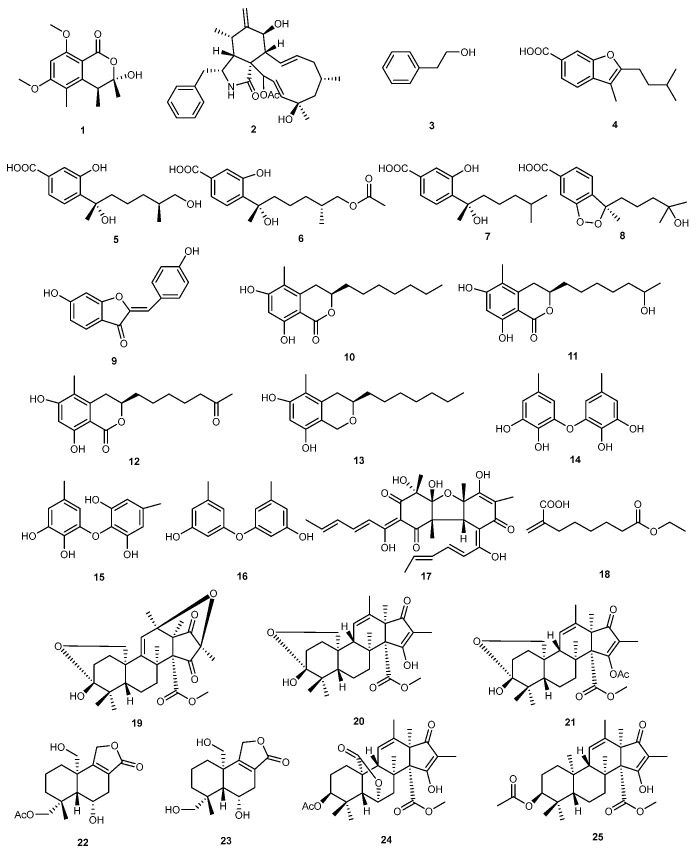
Chemical structures of compounds **1**–**25**.

**Figure 2 marinedrugs-23-00438-f002:**
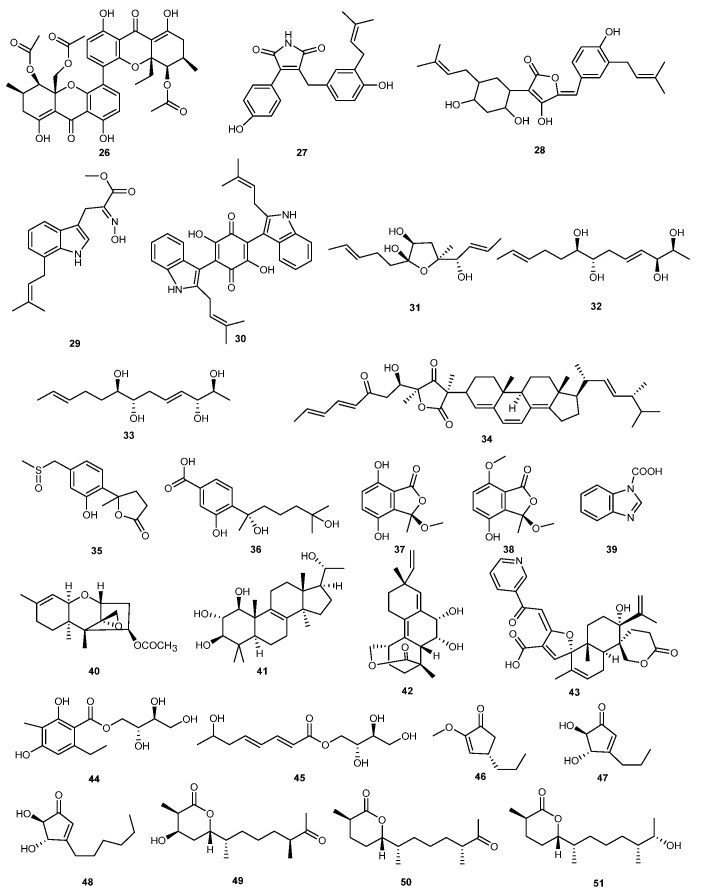
Chemical structures of compounds **26**–**51**.

**Figure 3 marinedrugs-23-00438-f003:**
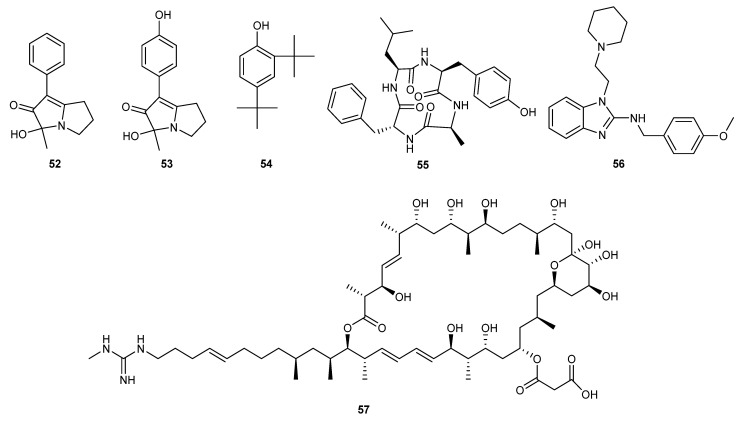
Chemical structures of compounds **52**–**57**.

**Figure 4 marinedrugs-23-00438-f004:**
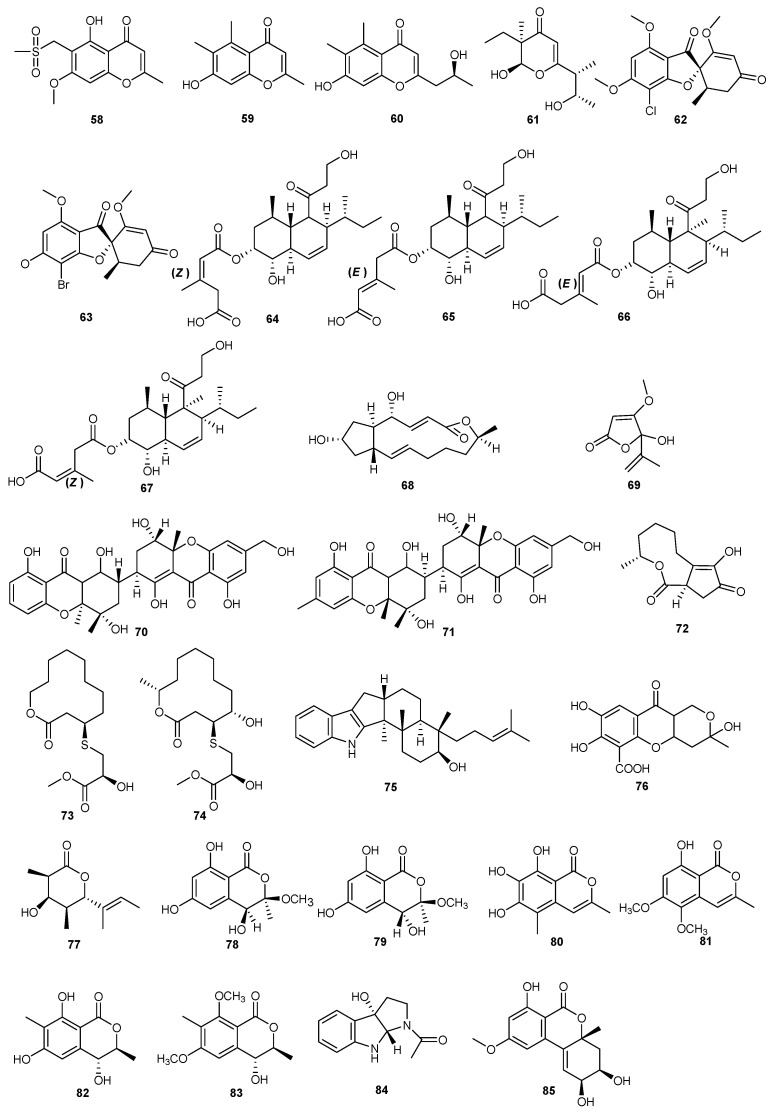
Chemical structures of compounds **58**–**85**.

**Figure 5 marinedrugs-23-00438-f005:**
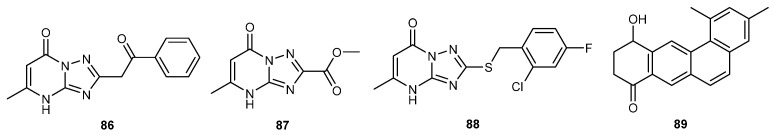
Chemical structures of compounds **86**–**89**.

**Figure 6 marinedrugs-23-00438-f006:**
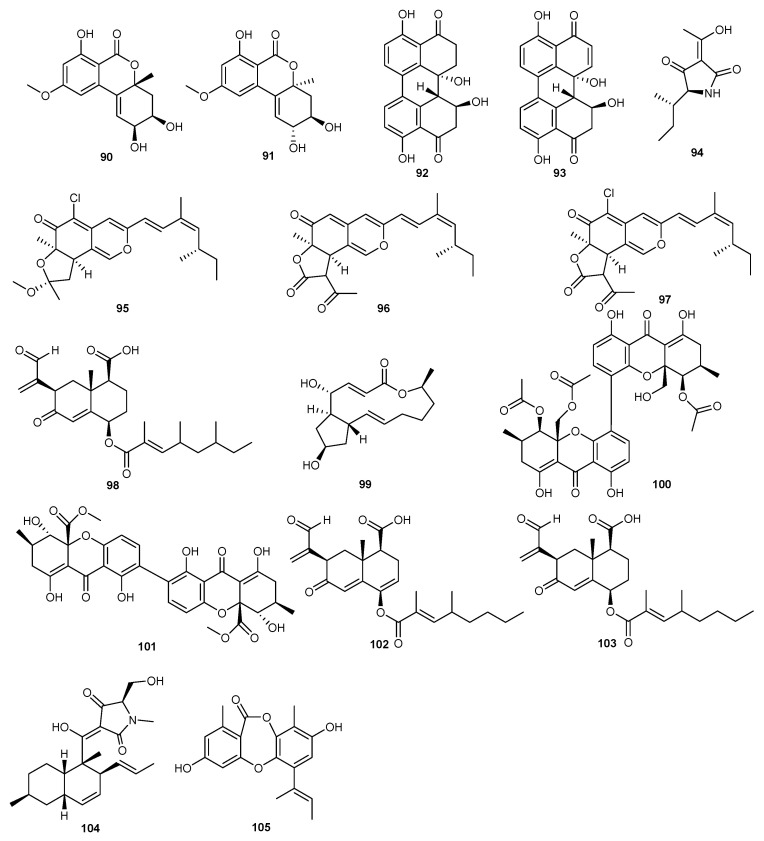
Chemical structures of compounds **90**–**105**.

**Figure 7 marinedrugs-23-00438-f007:**
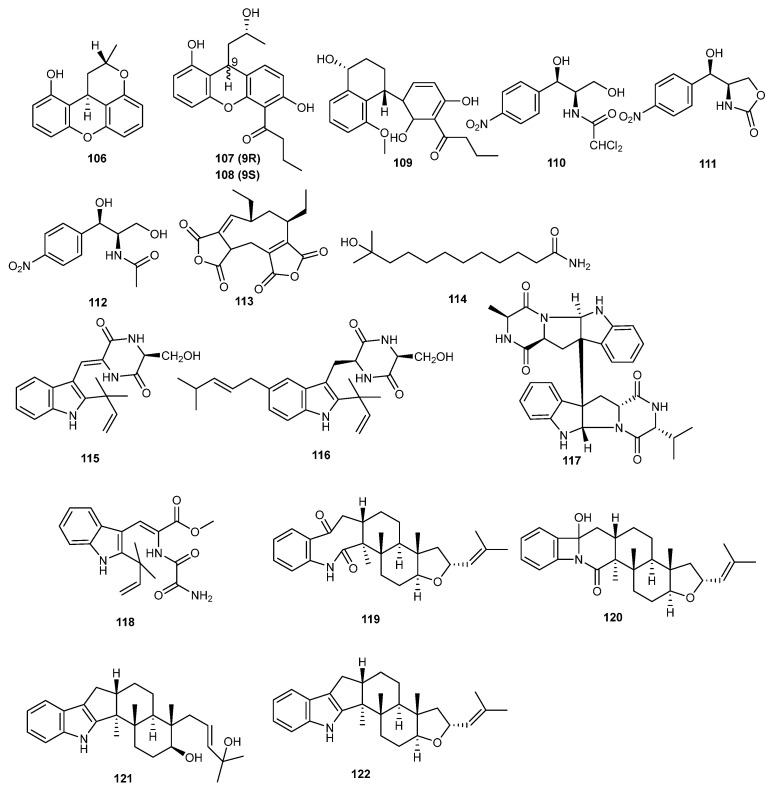
Chemical structures of compounds **106**–**122**.

**Figure 8 marinedrugs-23-00438-f008:**
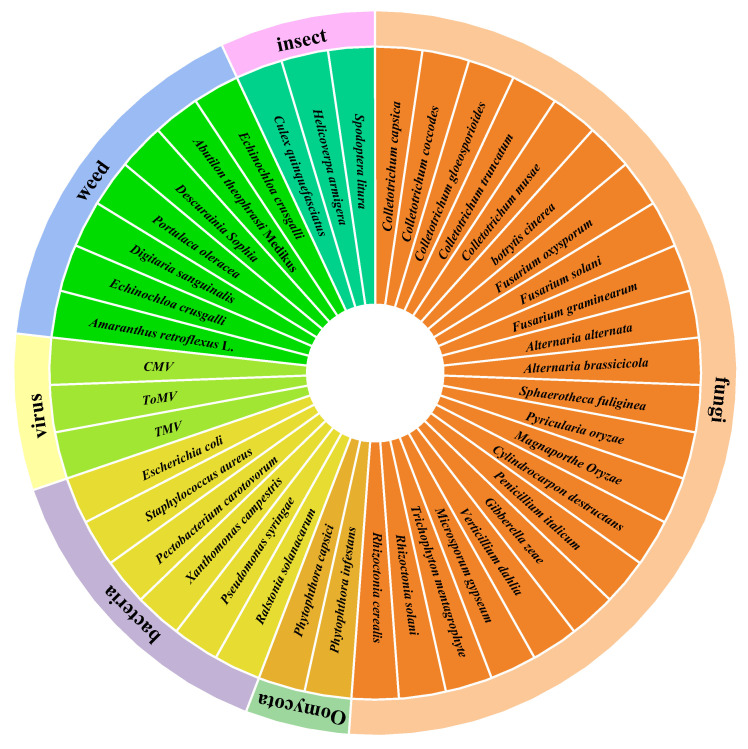
The antagonistic activity of secondary metabolites.

**Figure 9 marinedrugs-23-00438-f009:**
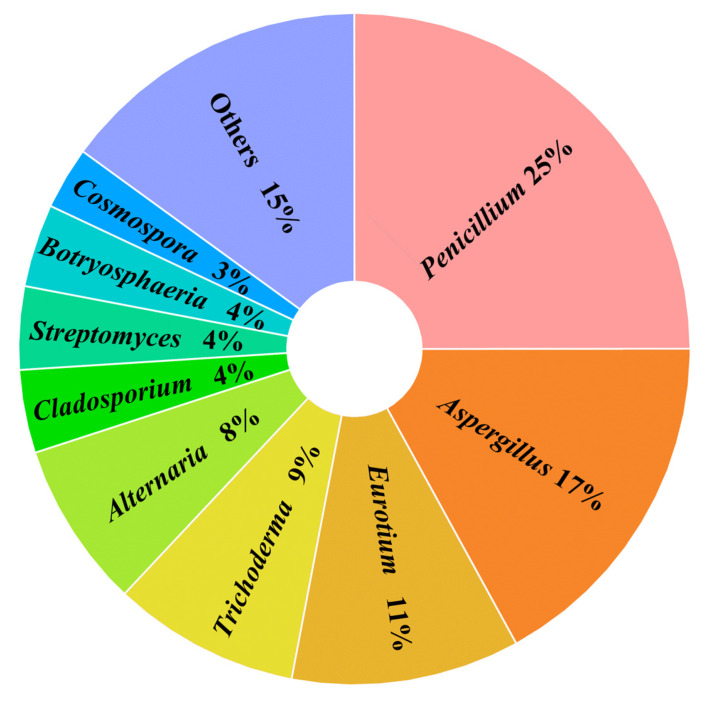
The proportion of secondary metabolites from fungal origin.

**Figure 10 marinedrugs-23-00438-f010:**
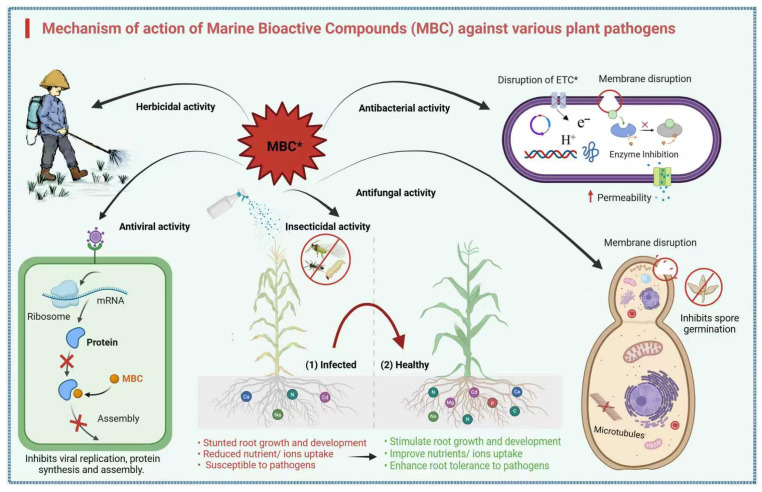
Mechanism of action of MBCs against plant pathogens and their beneficial effects on plant health. The figure illustrates the multifaceted roles of MBCs in combating various plant pathogens, including herbicides, viruses, insects, and fungi, through mechanisms such as disruption of the electron transport chain (ETC*), membrane disruption, enzyme inhibition, and inhibition of viral replication, protein synthesis, and assembly. Additionally, MBCs promote plant health by stimulating root growth and development, improving nutrient uptake, and enhancing root tolerance to pathogens. Key processes, such as microtubule interference, spore germination inhibition, and modulation of ribosomal activity, are also highlighted. The schematic provides a comprehensive overview of the dual functionality of MBCs in pathogen suppression and plant growth enhancement.

**Table 1 marinedrugs-23-00438-t001:** Secondary metabolites associated with marine origin.

No.	Compound	Producing Strain	Active	Ref.
**1**	cladosporinisochromanone	*Cladosporium* sp. DLT-5	-	[16]
**2**	cytochalasin H	*Cladosporium* sp. DLT-5	*Colletotrichum capsici*	[16]
**3**	2-phenylethanol	*Scheffersomyces spartinae* W9	*Botrytis cinerea*	[17]
**4**	aspergillusene B	*Aspergillus sydowii* LW09	*Ralstonia solanacarum*	[18]
**5**	(7S,11S)-(+)-12-hydroxysydonic acid	*Aspergillus sydowii* LW09	*Pseudomonas syringae*	[18]
**6**	expansol G	*Aspergillus sydowii* LW09	*Ralstonia solanacarum*	[18]
**7**	(S)-sydonic acid	*Aspergillus sydowii* LW09	*Fusarium oxysporum*	[18]
**8**	(−)-(R)-cyclo-hydroxysydonic acid	*Aspergillus sydowii* LW09	*Alternaria alternata*	[18]
**9**	aurone	*Penicillium* sp. CRM1540	*Botrytis cinerea*, *Sphaerotheca fuliginea*, *Pyricularia oryzae*	[20]
**10**	soudanone A	*Cosmospora* sp.	-	[21]
**11**	soudanone E	*Cosmospora* sp.	*Magnaporthe oryzae*, *Phytophthora infestans*	[21]
**12**	soudanone D	*Cosmospora* sp.	*Magnaporthe oryzae*, *Phytophthora infestans*	[21]
**13**	pseudoanguillosporins A	*Cosmospora* sp.	*Pseudomonas syringae*, *Xanthomonas campestris*, *Magnaporthe oryzae*, *Phytophthora infestans*	[21]
**14**	violaceol I	*Aspergillus tabacinus*	-	[22]
**15**	violaceol II	*Aspergillus tabacinus*	-	[22]
**16**	diorcinol	*Aspergillus tabacinus*	*Magnaporthe oryzae*, *Phytophthora infestans*, *Colletotrichum coccodes*	[22]
**17**	bisvertinolone	*Trichoderma longibrachiatum*	*Colletotrichum coccodes*, *Cylindrocarpon destructans*, *Magnaporthe oryzae*	[23]
**18**	8-Ethyl 2-methyleneoctanedioate	*Aspergillus* sp. SCAU150	*Fusarium solani* bio-80814	[24]
**19**	hemiacetalmeroterpenoids A	*Penicillium* sp. N-5	*Penicillium italicum*, *Colletrichum gloeosporioides*	[25]
**20**	hemiacetalmeroterpenoids B	*Penicillium* sp. N-5	-	[25]
**21**	hemiacetalmeroterpenoids C	*Penicillium* sp. N-5	-	[25]
**22**	astellolide Q	*Penicillium* sp. N-5	-	[25]
**23**	parasiticolide A	*Penicillium* sp. N-5	-	[25]
**24**	citreohybridone A	*Penicillium* sp. N-5	*Penicillium italicum*, *Colletrichum gloeosporioides*	[25]
**25**	andrastins B	*Penicillium* sp. N-5	*Penicillium italicum*, *Colletrichum gloeosporioides*	[25]
**26**	methyleurotinone	*Eurotium rubrum*	*Pectobacterium carotovorum*	[26]
**27**	asperteramide	*Aspergillus terreus* BCC5799	*Alternaria brassicicola*	[27]
**28**	aspulvinone	*Aspergillus terreus* BCC5799	*Alternaria brassicicola*	[27]
**29**	luteoride E	*Aspergillus terreus* BCC5799	*Alternaria brassicicola*	[27]
**30**	asterriquinone CT5	*Aspergillus terreus* BCC5799	*Alternaria brassicicola*	[27]
**31**	trichoderols B	*Trichoderma* sp. Z43	*Artermia salina*	[28]
**32**	trichoderols E	*Trichoderma* sp. Z43	*Artermia salina*	[28]
**33**	triharzianin B	*Trichoderma* sp. Z43	*Artermia salina*	[28]
**34**	tennessenoid A	*Asperigills* sp. 1022LEF	*Fusarium oxysporum*	[29]
**35**	methylsulfinyl-1-hydroxyboivinianin A	*Penicillium chrysogenum* LD-201810	-	[30]
**36**	hydroxysydonic acid	*Penicillium chrysogenum* LD-201810	*Botrytis cinerea*	[30]
**37**	chrysoalide A	*Penicillium chrysogenum* LD-201810	-	[30]
**38**	chrysoalide B	*Penicillium chrysogenum* LD-201810	-	[30]
**39**	phenazine-1-carboxylic acid	*Emericella* sp. SMA01	*Phytophthora capsici*, *Gibberella zeae*, *Verticillium dahliae*	[31]
**40**	trichodermin	*Trichoderma brevicompactum* A-DL-9-2	*Botrytis cinerea*, *Fusarium oxysporum*	[32]
**41**	polyhydroxy steroid	*Beauveria feline*	*Botrytis cinerea*	[33]
**42**	tricyclic diterpenoid	*Beauveria feline*	*Botrytis cinerea*	[33]
**43**	oxalicine C	*Penicillium chrysogenum* XNM-12	*Ralstonia solanacearum*	[34]
**44**	penicierythritols A	*Penicillium chrysogenum* XNM-12	-	[34]
**45**	penicierythritols B	*Penicillium chrysogenum* XNM-12	*Ralstonia solanacearum*, *Alternaria alternata*	[34]
**46**	asperterrein	*Paecilomyces lilacinus* EN-531	*Alternaria brassicae*	[35]
**47**	dihydroterrein	*Paecilomyces lilacinus* EN-531	*Alternaria brassicae*	[35]
**48**	terrein	*Paecilomyces lilacinus* EN-531	*Alternaria brassicae*	[35]
**49**	chermesiterpenoids A	*Penicillium chermesinum* EN-480	*Colletottichum gloeosporioides*	[36]
**50**	chermesiterpenoids B	*Penicillium chermesinum* EN-480	*Colletottichum gloeosporioides*	[36]
**51**	chermesiterpenoids C	*Penicillium chermesinum* EN-480	*Colletottichum gloeosporioides*	[36]
**52**	phenopyrrolizins A	*Micromonospora* sp. HU138	*Botrytis cinerea*	[38]
**53**	phenopyrrolizins B	*Micromonospora* sp. HU138	*Botrytis cinerea*	[38]
**54**	2, 4-di-tert butyl phenol	*Serratia marcescens* BKACT	*Fusarium* sp.	[39]
**55**	arthropeptide B	*Arthrobacter humicola* M9-1A	*Alternaria alternata*	[40]
**56**	benzimidazole	*Kocuria palustris* 19C38A	*Fusarium oxysporum*	[43]
**57**	niphimycin C	*Streptomyces yongxingensis* sp. nov. (JCM 34965)	*Fusarium oxysporum* f. sp. *cubense*	[46]
**58**	alternanone A	*Alternaria iridiaustralis*	*Botrytis cinerea*	[47]
**59**	alternanone B	*Alternaria iridiaustralis*	*Botrytis cinerea*	[47]
**60**	chaetosemin D	*Alternaria iridiaustralis*	*Botrytis cinerea*	[47]
**61**	alternanone C	*Alternaria iridiaustralis*	*BSotrytis cinerea*	[47]
**62**	( +)-6′-Hydroxygriseofulvin	*Nigrospora* sp. QQYB1	*Colletotrichum truncatum*, *Microsporum gypseum*, *Trichophyton mentagrophyte*	[48]
**63**	Spiro[benzofuran-2(3H),1′-[2]cyclohexene]-3,4′-dione, 7-bromo-2′,4,6-trimethoxy-6′-methyl- (9CI)	*Nigrospora* sp. QQYB1	*Colletotrichum truncatum*, *Microsporum gypseum*, *Trichophyton mentagrophyte*	[48]
**64**	2-Pentenedioic acid, 3-methyl-, 1-[(1S,2R,4R,4aS,5S,6S,8aR)-1,2,3,4,4a,5,6,8a-octahydro-1-hydroxy-5-(3-hydroxy-1-oxopropyl)-4,5-dimethyl-6-[(1R)-1-methylpropyl]-2-naphthalenyl] ester, (2E)- (9CI, ACI)	*Trichoderma lentiforme* ML-P8-2	*Penicillium italicum*	[49]
**65**	2-Pentenedioic acid, 3-methyl-, 1-[(1S,2R,4R,4aS,5S,6S,8aR)-1,2,3,4,4a,5,6,8a-octahydro-1-hydroxy-5-(3-hydroxy-1-oxopropyl)-4,5-dimethyl-6-[(1R)-1-methylpropyl]-2-naphthalenyl] ester, (2E)- (9CI, ACI)	*Trichoderma lentiforme* ML-P8-2	*Penicillium italicum*	[49]
**66**	tandyukisin C	*Trichoderma lentiforme* ML-P8-2	*Penicillium italicum*	[49]
**67**	tandyukisin G	*Trichoderma lentiforme* ML-P8-2	*Penicillium italicum*	[49]
**68**	brefeldin A	*Penicillium javanicum* HK1-23	*Rhizoctonia solani*, *Rhizoctonia cerealis*	[50]
**69**	penicillic acid	*Penicillium javanicum* HK1-23	*Rhizoctonia solani*, *Rhizoctonia cerealis*	[50]
**70**	aflaxanthones A	*Aspergillus flavus* QQYZ	*Fusarium oxysporum*, *Collettrichum musae*	[51]
**7** **1**	aflaxanthones B	*Aspergillus flavus* QQYZ	*Fusarium oxysporum*, *Collettrichum musae*	[51]
**7** **2**	cladocladosin A	*Cladosporium cladosporioides* MA-299	*Fusarium oxysporum* f. sp. *momodicae*	[52]
**7** **3**	thiocladospolides F	*Cladosporium cladosporioides* MA-299	*Fusarium oxysporum* f. sp. *momodicae*	[52]
**7** **4**	thiocladospolides G	*Cladosporium cladosporioides* MA-299	*Fusarium oxysporum* f. sp. *momodicae*	[52]
**7** **5**	emindole SB	*Penicillium javanicum* HK1-23	*Alternaria alternata*	[53]
**7** **6**	penialidin A	*Penicillium javanicum* HK1-23	*Alternaria alternata*	[53]
**7** **7**	penicoffeazine A	*Penicillium coffeae* MA-314	*Fusarium oxysporum* f. sp. *momordicae* nov. f., *Colletotrichum gloeosporioides*	[54]
**78**	penicoffrazins B	*Penicillium coffeae* MA-314	-	[54]
**79**	penicoffrazins C	*Penicillium coffeae* MA-314	-	[54]
**80**	botryospyrones A	*Botryosphaeria ramose* L29	*Fusarium oxysporum*	[55]
**8** **1**	botryospyrones B	*Botryosphaeria ramose* L29	*Fusarium oxysporum*, *Fusarium graminearum*	[55]
**8** **2**	botryospyrones C	*Botryosphaeria ramose* L29	*Fusarium oxysporum*, *Fusarium graminearum*	[55]
**8** **3**	botryospyrones D	*Botryosphaeria ramose* L29	*Fusarium oxysporum*, *Fusarium graminearum*	[55]
**8** **4**	(3aS, 8aS)-1-acetyl-1, 2, 3, 3a, 8, 8a-hexahydropyrrolo [2,3b] indol-3a-ol	*Botryosphaeria ramose* L29	*Fusarium oxysporum*, *Fusarium graminearum*	[55]
**8** **5**	(+)-(2S,3R,4aR)-altenuene	*Alternaria* sp. (P8)	*Alternaria brassicicola*	[56]
**8** **6**	essramycin	*Streptomyces* sp. Merv8102	TMV	[64]
**8** **7**	Methyl 1,7-dihydro-5-methyl-7-oxo [1,2,4]triazolo [1,5-a]pyrimidine-2-carboxylate	*Streptomyces* sp. Merv8102	TMV	[64]
**88**	[1,2,4]Triazolo [1,5-a]pyrimidin-7(1H)-one, 2-[[(2-chloro-4-fluorophenyl)methyl]thio]-5-methyl- (ACI)	*Streptomyces* sp. Merv8102	TMV	[64]
**89**	acterophenone A	*Streptomyces* sp. KCB32	TMV, ToMV, CMV	[65]
**90**	(+)-(2S,3R,4aR)-altenuene	*Alternaria* sp. (P8)	*Amaranthus retroflexus* L., *Alternaria brassicicola*	[56]
**9** **1**	(+)-isoaltenuene	*Alternaria* sp. (P8)	*Amaranthus retroflexus* L.	[56]
**9** **2**	stemphyperylenol	*Alternaria* sp. (P8)	*Amaranthus retroflexus* L.	[56]
**9** **3**	alterperylenol	*Alternaria* sp. (P8)	*Amaranthus retroflexus* L.	[56]
**9** **4**	alkalodi	*Alternaria iridiaustralis*	*Echinochloa crusgalli*, *Digitaria sanguinalis*, *Portulaca oleracea*, *Descurainia sophia*	[47]
**9** **5**	isochromophilone H	*Penicillum sclerotiorum* HY5	*Amaranthus retroflexus* L.	[67]
**9** **6**	ochlephilone	*Penicillum sclerotiorum* HY5	*Amaranthus retroflexus* L., *Abutilon theophrasti* Medikus	[67]
**9** **7**	isochromophilone I	*Penicillum sclerotiorum* HY5	*Amaranthus retroflexus* L., *Abutilon theophrasti* Medikus	[67]
**98**	integric acid A	*Eurotium rubrum*	*Echinochloa crus-galli*	[26]
**99**	brifeldin A	*Eurotium rubrum*	*Echinochloa crus-galli*	[26]
**10** **0**	[4,4′-Bi-9H-xanthene]-9,9′-dione, 5,5′-bis(acetyloxy)-10a,10′a-bis[(acetyloxy)methyl]-5,5′,6,6′,7,7′,10a,10′a-octahydro-1,1′,8,8′-tetrahydroxy-6,6′-dimethyl-, (4R,5S,5′S,6S,6′S,10aS,10′aS)- (9CI, ACI)	*Eurotium rubrum*	*Echinochloa crus-galli*	[26]
**10** **1**	secalonic acid D	*Eurotium rubrum*	*Echinochloa crus-galli*	[26]
**10** **2**	eremoxylarin B	*Eurotium rubrum*	*Echinochloa crus-galli*	[26]
**10** **3**	integric acid A	*Eurotium rubrum*	*Echinochloa crus-galli*	[26]
**10** **4**	equisetin (EQ)	*Eurotium rubrum*	*Echinochloa crus-galli*	[26]
**10** **5**	unguinol	*Ianthella reticulate*	pyruvate phosphate dikinase	[68]
**10** **6**	penicixanthenes A	*Pencillium* sp. JY246	*Culex quinquefasciatus* larvae	[73]
**10** **7**	penicixanthenes B	*Pencillium* sp. JY246	*Helicoverpa armigera* Hubner larvae	[73]
**1** **08**	penicixanthenes C	*Pencillium* sp. JY246	*Helicoverpa armigera* Hubner larvae, *Culex quinquefasciatus* larvae	[73]
**1** **09**	penicixanthenes D	*Pencillium* sp. JY246	*Culex quinquefasciatus* larvae	[73]
**1** **10**	synthomycin	*Acremonium vitellinum*	*Helicoverpa armigera*	[74]
**11** **1**	(4R)-4-[(R)-Hydroxy(4-nitrophenyl)methyl]-2-oxazolidinone	*Acremonium vitellinum*	-	[74]
**11** **2**	N-[(1R,2R)-2-Hydroxy-1-(hydroxymethyl)-2-(4-nitrophenyl)ethyl]acetamide	*Acremonium vitellinum*	-	[74]
**11** **3**	aspergide	*Aspergillus fumigatus* JRJ111048	*Spodoptera litura*	[75]
**11** **4**	11-methyl-11-hydroxyldodecanoic acid amide	*Aspergillus fumigatus* JRJ111048	-	[75]
**11** **5**	cristatumins A	*Eurotium cristatum* EN-220	*E. coli*, *Staphylococcus aureus*	[76]
**11** **6**	cristatumins B	*Eurotium cristatum* EN-220		[76]
**11** **7**	cristatumins C	*Eurotium cristatum* EN-220	-	[76]
**1** **18**	cristatumins D	*Eurotium cristatum* EN-220	*S. aureus*, *Staphylococcus aureus*	[76]
**1** **19**	asporyzin A	*Aspergillus oryzae*		[77]
**1** **20**	asporyzin B	*Aspergillus oryzae*		[77]
**12** **1**	asporyzin C	*Aspergillus oryzae*	*Brine shrimp*	[77]
**12** **2**	JBIR-03	*Aspergillus oryzae*		[77]

## Data Availability

Not applicable.

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
