# Peer review of "Natural Products from Marine Microorganisms with Agricultural Applications"

_marinedrugs, 2025, doi:10.3390/md23110438_

Round 1

Reviewer 1 Report

Comments and Suggestions for Authors

This paper summarizes 122 microbial secondary metabolites with agricultural activities. 71 compounds exhibit antibacterial activity, 4 antiviral, 16 herbicidal, and 15 insecticidal. The compounds demonstrate a broad spectrum of activity against multiple genera and species of agricultural pathogens, weeds, and pests.  Natural products from marine microorganisms have shown inhibitory effects against a wide range of pathogens. Cladosporium sp. produced a novel isochromanone that exhibited excellent antifungal activity against Colletotrichum sp. VOCs produced by Scheffersomyces spartinae W9 inhibited the mycelial growth of Botrytis cinerea and spore germination.

1.- The manuscript summarizes various bioactive compounds but does not extensively elaborate on their mechanisms of pesticidal activity, which is crucial for understanding their potential application and development.

2.- While many compounds demonstrate activity in vitro (e.g., MIC values, LC50), there is limited information on their efficacy in field conditions and potential toxicity to non-target organisms, including humans, which is essential for practical application.

3.- The review presents numerous compounds with bioactivities but does not discuss SAR insights that could guide future compound optimization.

4.- The review encompasses many compounds but may benefit from more detailed case studies or examples demonstrating successful development or application of marine-derived pesticides.

Reviewer 2 Report

Comments and Suggestions for Authors

Line 118: compound 9 is a derivative of aurone!

Line 119: what is “herboxidiene”? The chemical structure does not appear.

Lines 130-131: compounds 14, 15 and 16 are diphenyl ethers, the name benzyl ethers is incorrect.

Line 188 and fig. 2: the structure of compound 38 is wrong, or the chemical name is wrong (the structure in the figure is of benzimidazole derivative).

The structures of compounds fengycin A (line 253), surfactin (line 254) and cyclic lipopeptide C17-fenngycin B (line 267) do not appear in the manuscript.

Lines 289-290: only compounds 62 and 63 are derivatives of griseofulvin.

Attention: Aflaxanthone A (line 307) and Aflaxanthone B (line 309).

Line 430: attention to the compounds, are they really nitrides?

Line 433 and fig. 7: compound 109 does not have a xanthene structure!

Line 450-451: the correct name for compound 114 is 11-hydrox-11-methyldodecanoic acid amide.

Lines 458-459: compound 119 and compound 120 are not indole derivatives.

The text does not mention the figures and table in the manuscript.

In chapter 3.1, a series of bioactive substances are mentioned that are not described in the previous chapters (see the attached document).

Attention to some acronyms that must be explained at the first appearance in the text.

Round 2

Reviewer 1 Report

Comments and Suggestions for Authors All the changes requested from the authors have been made. Therefore, I appreciate the improvement to their work. I have no further comments.